# Features and Sustainable Design of Firefighting Safety Footwear for Fire Extinguishing and Rescue Operations

Marianna Tomaskova and Jozef Krajňák *

Faculty of Mechanical Engineering, Technical University of Košice, Letná 9 Street, 04200 Košice, Slovakia; marianna.tomaskova@tuke.sk
* Correspondence: jozef.krajnak@tuke.sk; Tel.: +421-602-2373

**Abstract:** Firefighters are regularly exposed to risk of injuries as a result of their intervention activities connected with hazards from fire and explosion, as well as due to carrying heavy personal working equipment and injured victims from accidents. Another hazardous factor is working under unfavorable weather conditions and also moving on slippery or bumpy surfaces. Employers provide personal protective work equipment to employees if a hazard cannot be eliminated or reduced by technical means, such as means of collective protection or methods and forms of work organization. Personal Protective Equipment (PPE) should provide effective protection against existing and foreseeable hazards and should not in itself create a greater risk. It should be adapted to the existing and predictable working conditions and working environment at the work site, meet the criteria of ergonomics and the health condition of an employee as well as be suitable and adapted to the wearer's body so that, if possible, there is no risk of harm to the employee's health. The aim of this Special Issue is to explore the limits of sustainable implementation of additive technologies within current manufacturing practices and current requirements for personal protective equipment for firefighters. More precisely, the goal of this special article is to show new ideas in firefighting footwear such as a quick donning and doffing system as well as various other improvements and sustainable design of firefighting footwear. The aim is to present new ideas and concepts, the latest advances, and technical tools supporting the sustainable use of protective firefighting footwear. Special attention will be paid to standards that ensure the highest standard and quality.

**Keywords:** sustainable design; fire; PPE; firefighting boots; sustainable materials; test methods; hazard

## 1. Introduction

The profession of firefighter belongs to the category of riskiest professions. There were an increasing number of large forest fires recorded during the last period. This negative fact is directly connected with rapid climatic changes. The firefighting interventions in the case of forest fires are very difficult and dangerous. In this context, the personal protective equipment worn by the firefighters is vital. This article is focused on firefighting footwear with regard to safety and utilisation.

A report from the IPCC (Intergovernmental Panel on Climate Change) [1] highlighted a relevant fact that, compared to the years 1986–2005, the global surface temperature, measured over the next several decades, will be increased by 0.3 °C to 4.8 °C according to various climatic scenarios. At the same time, the extreme values of cold and hot temperatures will occur more frequently across all areas. It is a well-known fact that the climate has a strong impact on forest growth [2,3]. Therefore, the strong effects of climatic changes on the forests must be considered seriously. This can cause an increase in the number of forest fires. These fires will be extinguished by firefighters, so their safety is important.

Nowadays, wildfires belong among the most hazardous and most catastrophic natural disasters. Serious wildfires were recorded around the world during the last several years [4,5]. Forest fires occur all over the world. Therefore, the safety and security of

firefighters is very important. The United States, Canada and Australia are classified at the top of the "black-list" of the countries affected by the wildfires, taking into consideration their climate, topographical conditions and vegetation [6,7]. Australia has a long history of large-scale wildfires that occur during periods of high temperatures and other extreme meteorological conditions. No wonder then that the Australian continent is the driest continent. There can be mentioned, for example, the wildfires in the states of Victoria and New South Wales, 1939, in Tasmania, 1967, in Victoria, 1977 and the "Ash Wednesday" fires in Victoria and South Australia, 1983 [8]. The most recent serious wildfires are the devastating "Black Saturday" fires in Victoria, 2009 and those in New South Wales, 2019–2020 [8]. These mentioned wildfires resulted in thousands of destroyed houses and other buildings, hundreds of fatalities and material losses expressed in millions of dollars. The greatest tragedies of the Black Saturday wildfires in 2009 were that they destroyed more than 1800 homes, burned about 270,000 ha of land and caused the deaths of 173 people [9]. Other catastrophic wildfires in New South Wales, during the wildfire season 2019–2020 in Australia, damaged 3000 homes, claimed 34 human lives and caused material losses over $110 billion [8,9]. It is evident that the wildfires in Australia are causing terrible social and economic impacts. In the last few years, the forest fires have developed with great intensity and speed of spread, causing an increasing number of problems in the application of existing extinguishing systems. The large forest fires that occurred in the recent past required innovative firefighting strategies and tactics, new firefighting equipment, firefighting tools and personal protective means for the fire brigade members. Extinguishing the forest fires in mountain terrain, which is essentially inaccessible, requires the total deployment of the firefighting units and utilisation of all their modules. The forest fires are characterized by a very fast spread over large areas. Liquidation of such a fire is lengthy, while the possibility of its re-ignition, originating from hidden sources of fire, cannot be neglected. Therefore, it is necessary to control the whole fire area. In action, the forest firefighters are moving towards the source of fire to suppress it and to minimize damages to the environment, workspaces and homes, and to protect potential victims such as humans and wildlife. The role of the forest firefighters is becoming ever more important with regard to the increasing number of forest fires that are occurring in the summer months characterised by periods of unbreathable air and "to stay home" orders [10]. For the forest firefighters, footwear suitable for firefighting intervention, i.e., firefighter boots, is extremely important. One of the effects of a forest fire on the soil is it overheating and for this reason it is very important to choose suitable footwear for the fire brigade members, namely for the forest firefighters. During a forest fire, the temperature just above the soil surface can exceed 700 °C (Daubenmire 1968 in Pyne et al., 1996) [11]. Many times during forest fires, temperatures exceeding 800 °C in the surface layers of the soil have been recorded. (DeBano et al., 1979 and Pyne et al., 1996). The temperature deeper in the soil is usually lower than at the surface. In the case of intensive fires in green dense bushes, the maximum temperature in a depth of about 2.5 cm does not exceed 200 °C, but the temperature just below the surface layer reaches up to 275 °C (Beadle 1940 in Pyne et al., 1996). That is why the application of proper protective footwear for the personnel involved is an unavoidable requirement [11].

Weather and flammability of various materials play substantial roles in the occurrence of wildfires. The long-time, durable high temperatures and dry weather as well as high wind speeds accelerate the loss of water from the air and soil [12]. Subsequently, these factors significantly increase the danger of wildfire occurrence [13]. The most important weather parameters, for example wind speed, surface temperature, relative humidity and rainfall, enable us to estimate the danger of a wildfire [14–16]. These parameters are monitored by meteorological stations [17] or they can be simulated by means of a specific climate analysis model [18,19].

In addition to the above-mentioned facts, Nathan Phelps [20] investigated studies concerning assessment of wildfire danger that were realised during the previous 20 years and he proposed that a complex consideration of terrain and ignition source is very helpful and effective for an improved assessment of the wildfire danger.

The firefighters are exposed to dangerous thermal stress situations, which can cause severe burns and also death. To avoid such negative situations, the firefighter suit plays a very important role [21]. Hence, improvements in firefighting working clothes are necessary [22–28]. It is necessary to pay special attention to firefighter's footwear, to its safety and also to its constant improvement, so that it could be sustainable and usable in practice for a long time.

What makes a good firefighting boot [29]?

Firefighters are aware of how important it is to protect their feet. Long hours on duty can take their toll on them, so wearing comfortable and safe footwear is vital. What makes good firefighting boots? They must provide support, protection and comfort. From climbing rough terrain and debris, to the hazards of falling objects and extreme heat and cold, firefighting boots must cope with a lot. They must be robust enough not to be punctured and strong enough to walk on molten metal and withstand the heat. Firefighters spend a lot of time on their feet, so comfort comes equal first with quality [30–34].

Although modifications to the design of boots increase their safety, they do not always provide optimal performance in a range of working conditions. Understanding the impact of these differences can help develop better design improvements to ensure the safe and effective performance of tasks. The assessed biomechanical variables include, e.g., gait, stability, muscle and slip activity, but also physiological variables such as energy expenditure, heart rate, pain perception and oxygen consumption. As a medium between feet and contact surface, boots are a very important extrinsic factor in postural stability, gait and risk of falling. Based on many studies investigating the impact of three types of safety boots (steel-toed boots, tactical footwear and non-slip footwear) on the human biomechanics and physiology of postural control, it can be said that the footwear provides several positive design features, which affect the postural control [35].

Study [36] investigated the influence of the varying stiffness of boots on the kinematics and kinetics of gait. Stiffness of boot was examined by measuring the dependence force deformation. The model of a foot, inserted into the boot shoe, was pushed by a special robot. The gait analysis was realised on nine neurologically intact subjects during walking with two different shoes both with backpack and without backpack. The measured differences were statistically tested by means of the statistical software product. The obtained results pointed to differences in stiffness of boot shaft and in instep. The boot with softer shaft allowed for a more flexible motion in the ankle joint, resulting in greater energy production of the ankle joint during rebound, as well as in increased stride length and walking speed. The backpack affected kinematics of the pelvis and trunk. The performed study presented the impact of boot stiffness on the parameters of biomechanical gait and on their importance for rebound, which manufacturers should consider when optimizing boot performance. The aim is to present new ideas, concepts, latest advances and technical tools supporting the sustainable use of protective firefighting footwear. Special attention will be paid to technical standards that ensure the highest standard and quality.

Firefighters are often exposed to high temperatures and heat flows due to high levels of radiation coming from fire during firefighting. Therefore, personal protection is extremely important for firefighters to ensure their protection during firefighting activities.

It is necessary to emphasize the fact that this article is focused on the firefighting footwear determined for the firefighters who are working outdoors in terrain, usually in hard-to-reach places, e.g., in the forest, where they have to carry various firefighting equipment, such as water pumps or aggregators and similar special technical devices [37–39] to the place of intervention. Such activities are physically very demanding.

General requirements for safety and health protection according to Regulation (EU) 2016/425 of the European Parliament and European Council from 9 March 2016 on Personal Protective Equipment (PPE) are compulsory.

The main purpose of this article is to present new ideas, concepts, latest advances and technical tools supporting the sustainable use of protective firefighting footwear. Special attention will be paid to standards that ensure the highest standard and quality.

## 2. General Requirements Relating to All Kinds of PPE

Members of the Fire-Rescue service work under dangerous conditions, risking their own lives to save others. This demanding profession requires cutting-edge technology and all equipment in a flawless technical condition—including footwear, of course. It should combine sufficient comfort and technical features that will keep firefighters safe.

General requirements applicable to all PPE are given in Figure 1 and classification of the footwear for firefighters according to the valid Slovak technical standard STN EN 1509 [26] is presented in the following Table 1.

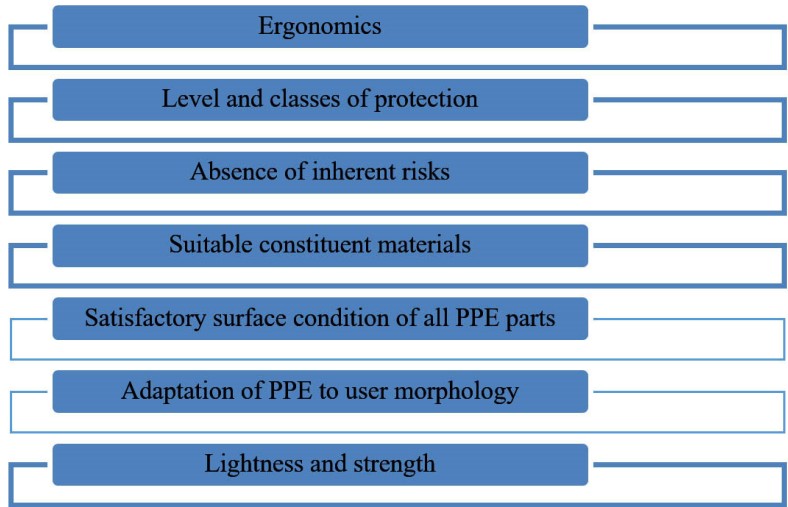

**Figure 1.** Requirements for PPE.

**Table 1.** Footwear for firefighters classified according to STN EN 15090 standard [26].

| Classification | Description |
|---|---|
| Class I | Footwear produced from leather or from other kind of material, except all-polymer or all-rubber footwear |
| Class II | All-polymer footwear (entirely moulded) or all-rubber footwear (entirely vulcanised) |

The firefighting boots described in this article are made from a special rubber compound that can withstand extremely high temperatures and provide puncture resistance. It is a type of special work footwear that can handle the most demanding conditions and tense situations that firefighters find themselves in. The quality of footwear depends on the technologies used. They are the result of many years of development and testing. Footwear for firefighters, its testing and risks are described in the Slovak technical standard STN EN 15090. The standard specifies the minimum requirements for properties and test methods for footwear for firefighters. The aim of the risk assessment is to determine whether footwear is suitable for use in emergency situations.

## 3. Test Methods for Firefighting Footwear

Types of firefighting footwear are divided into three main groups according to the STN EN 15090. These basic types of firefighting footwear and their characteristics summarised in Figure 2:

Type 1—suitable for general rescue, firefighting involving forest fires, agricultural crops or agricultural land.
Type 2—suitable for general rescue operations, fire extinguishing in buildings, in closed structures involved in fire. Type 2 includes all risks of Type 1.

Type 3—suitable for hazardous material emergencies involving the release of dangerous chemicals, property conservation in aircraft, buildings, vehicles, etc.

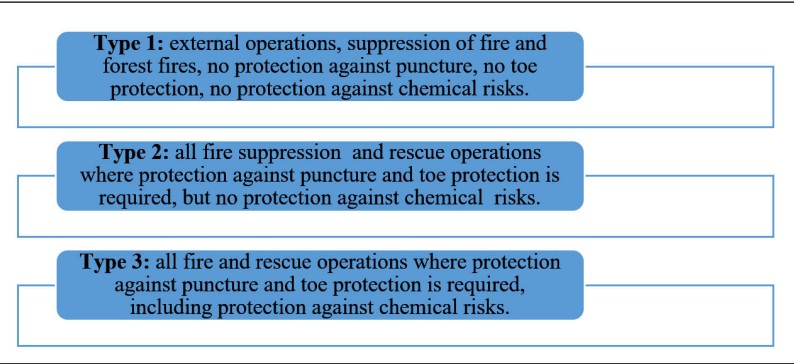

**Figure 2.** Types of firefighter footwear according to STN EN 15090.

Test methods for firefighting footwear according to STN EN 15090 can be divided into the following:

- Insulation against heat;
- Radiant heat;
- Flame resistant test;
- Pressure resistance of toe cap;
- Test of zipper.

### 3.1. Testing of Footwear for Flame Resistance

The EN ISO 15025:2002 standard [40] specifies the minimum requirements and test methods for three types of footwear intended for the firefighters in firefighting, namely in the normal rescue operations, in the fire rescue operations and in the hazardous materials accidents.

The above-mentioned standard defines various footwear tests. Only some of them are selected and described in this article. The pressure resistance test of the toe cap is tested according to the EN ISO 20344:2011 standard [41]. There are, in the case of the protective toe cap, specified requirements for the internal length of the reinforcements, for the impact resistance, pressure resistance and also for the corrosion resistance of the metal reinforcement.

Insulation against heat is evaluated according to the EN ISO 20344:2011 standard, where two test bodies from all different material combinations must be tested against radiation heat. The tested samples are taken from the uppers of one pair of shoes. The test bodies are tested according to the method specified in the EN ISO 6942:2002 [42] standard at a heat flow intensity of 20 kW/m$^2$ by exposing the outer surface to radiation heat for 40 s. One of the methods of testing firefighting footwear is the Flame Resistance Test. The burner is placed on horizontal surface, whereas the burner with flame is situated in a vertical position according to EN ISO 15025:2002 [40]. Such test arrangement is illustrated in Figure 3.

The part of the footwear to be tested is clamped in such a way that the minimal distance between the top of burner and the footwear is 17 ± 1 mm. The value of the angle between surface of the tested sample and the horizontal plane is 45 ± 5°. The sample carrier has a square opening of size (50 ± 1 mm) × (50 ± 1 mm) to apply the flame.

The burner is moved away from the sample, and it is ignited and preheated for two minutes, then the flame is adjusted to 35 ± 2 mm in height in accordance with EN ISO 15025. Then the burner is repositioned, and the flame is applied for 10 ± 1 s to the designated area. After that the flame is removed and the after-flame is measured. This procedure is applied for materials applied in production of footwear, closing mechanisms and external seams. Further, the rigidity of toe cap compressed with a load and the zipper are tested.

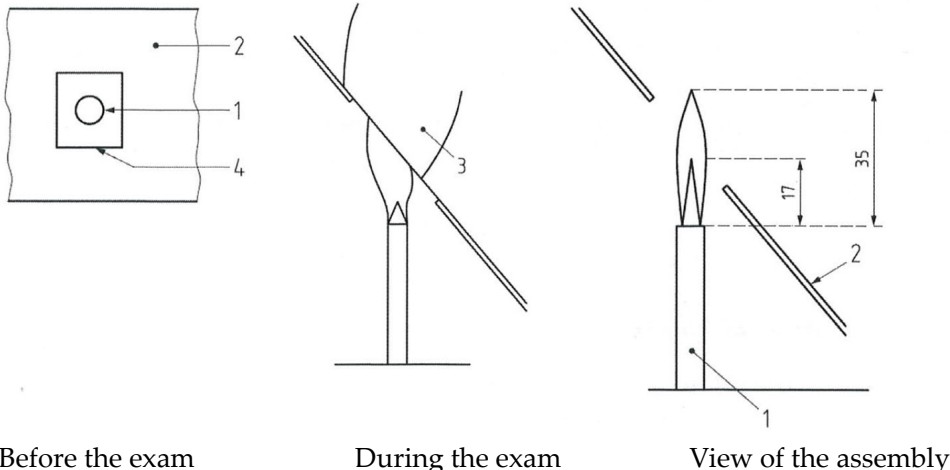

Before the exam | During the exam | View of the assembly

**Figure 3.** Apparatus for flame resistance test, 1—burner, 2—sample carrier, 3—footwear to be tested, 4—opening for application of flame (STN EN 15090).

*3.2. Test of Zipper*

In this test, the puller attachment strength and lateral strength are determined. The tested part is mounted in the jaw of the tester so that there is a minimum length of 25 mm of closed chain on both sides of the jaws at disposal. The jaws should be positioned 3 mm from the chain. This described arrangement is shown in Figure 4. The machine is set in operation and the force to induce failure is measured, at least three test pieces shall be tested, and the results recorded.

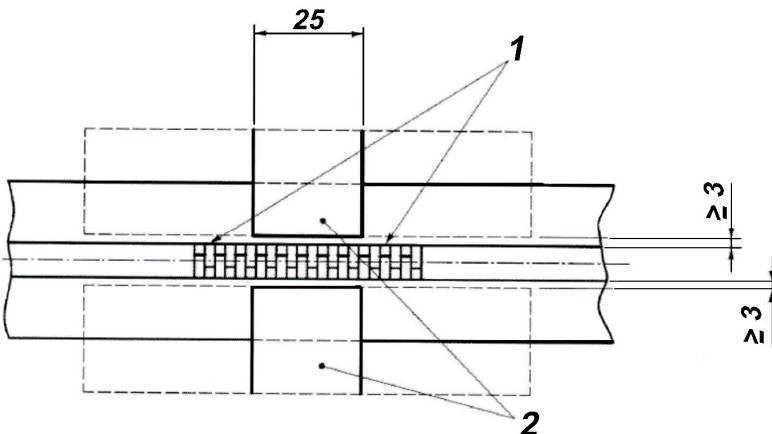

**Figure 4.** Test of footwear zipper, 1—at least 25 mm of the closed zipper on either side of the jaws, 2—jaws.

## 4. Marking of Firefighting Footwear

Each pair boots for firefighters shall be clearly and permanently marked, offering the following important information: size of boots, identification of the producer, designation of the given product, production year, standard number, marking symbols as shown in Figure 5. The pictogram sign in size 30 × 30 mm shall be affixed in a visible place on the outer side of boots (STN EN 15090).

Table 2 shows the agreed designation of the firefighter's footwear. Namely, there are three main types of footwear presented, which have their own symbols, as it can be seen in this table.

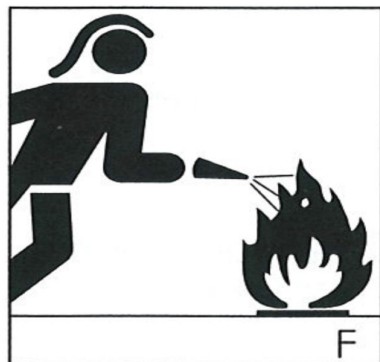

**Figure 5.** Pictogram designating type and footwear protection for firefighters.

**Table 2.** Another designation symbols according to STN EN 15090 standard.

| Type of Footwear | Symbol | Designated Properties |
|---|---|---|
| Type 1 | F1A | All the normative requirements are defined in the STN EN 15090 standard and requirements for antistatic properties |
| | F1PA | Requirements for puncture resistance and antistatic properties |
| | F1I | Requirements for electrical insulation properties |
| | F1PI | Requirements for puncture resistance and to electrical insulation properties |
| Type 2 | F2A | All the normative requirements are defined in the STN EN 15090 standard and requirements for antistatic properties |
| | F2I | Requirements for electrical insulation properties |
| Type 3 | F3A | All the normative requirements are defined in the STN EN 15090 standard and requirements for antistatic properties |
| | F3I | Requirements for electrical insulation properties |

## 5. Assessment of Risks

Types of appropriate PPE should be selected based on the assessment of a specific source of hazard. First, identification of threats, their evaluation and selection of specific requirements for the properties that minimize or reduce safety hazards are carried out according to STN EN 15090 [26].

General approach to risk assessment:

1.  Identification of risk

A list of possible risks associated with Fire department activities is developed based on records of accidents, diseases, injuries, etc.

2.  Assessment of risk

Risk assessment asks the questions: What is the level of occurrence and seriousness of the event?

How often do events occur, or how likely are they to occur?

What are possible consequences of their occurrence?

3.  Management of risk

After identification and assessment of risks it is necessary to record and adequately control each of them.

Specification of suitable protective equipment should be part of any comprehensive safety program that involves standard operational methods, procedures, training and inspections.

*Criteria for Risk Assessment*

A set of firefighting equipment also includes safety footwear that provides necessary protection when working in a potentially dangerous environment. In addition to high temperatures, firefighters are exposed to other dangers and face serious risks on the job, e.g., slippery surfaces, difficult terrain, flooded areas, sharp objects, etc. Taking all that into account, firefighting boots have to ensure a reliable traction, proper fit and ability of flexible movement. Thus, they must be produced from such materials which provide high resistance and protection, but at the same time are lightweight and comfortable.

Firefighting footwear performance standards include:

- Thermal resistance;
- Anti-corrosion resistance;
- Resistance to puncture, cut and abrasion;
- Resistance to conductive heat (limited inner sole temperature);
- Anti-slip resistance;
- Attachment strength (closure, cover cuffs and lacing);
- Anti-flame resistance;
- Thermal resistance of sewing thread.

When assessing the risk, it is important to consider the type and the incident command system. The level of discipline and coordination of firefighters at the site of the rescue operation may affect the risk of injury. For firefighters whose responsibilities are clearly defined and very well managed the probability to be injured is lower compared with firefighters who act independently or in a less coordinated manner.

Attention should be paid to alleviating heat strain on firefighters due to prolonged wearing of PPE when attending a fire incident and other related activities. Heat strain and other related stresses are one of the frequent causes of firefighters' fatalities and injuries. The most important factors affecting the firefighters during intervention, according to STN EN 15090, are summarised in Figure 6.

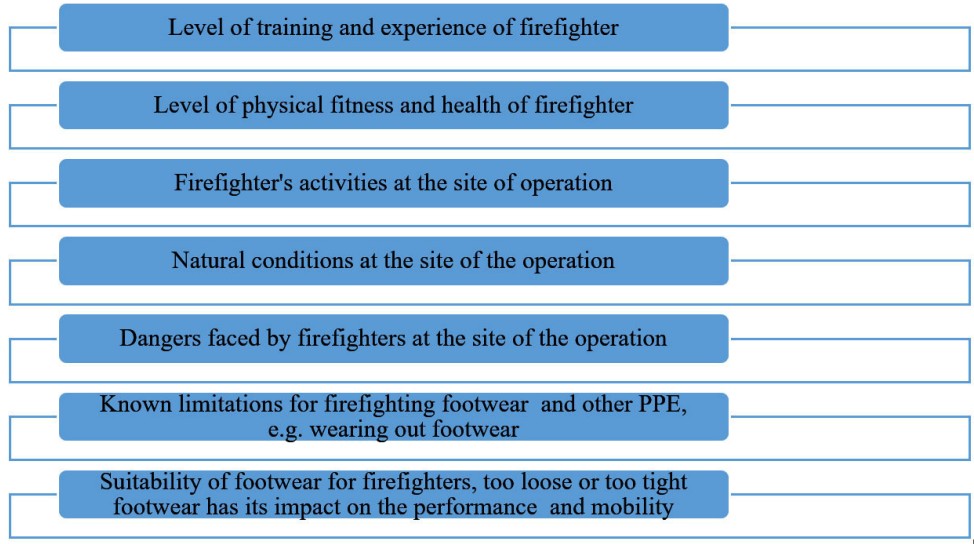

**Figure 6.** Factors affecting firefighters at intervention according to STN EN 15090 [26].

## 6. Example of Footwear Suitability Assessment by Testing Thermal Properties in the Laboratory

There are, as illustrated in the next six pictures (Figure 7), different criteria for judging of firefighter footwear in laboratory conditions.

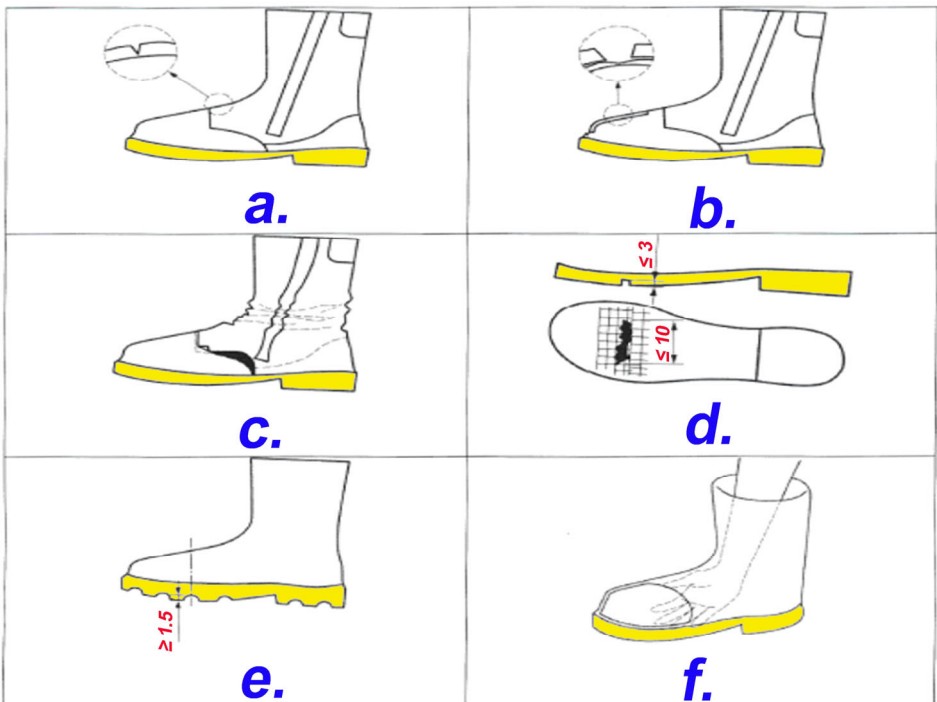

**Figure 7.** Criteria for the assessment of the state of firefighters' boots, Explanatory notes: (**a**). Resistance of the shoe instep, (**b**). Resistance of the shoe vamp, (**c**). Resistance to disruption of seams, (**d**). Outsole resistant to perforation, cut, puncture, (**e**). Outsole anti-slippery properties, (**f**). Impemeability.

*6.1. Importance of Protective Equipment during Firefighters' Interventions*

Personal protective equipment should protect against hazards that are summarized in Table 3, according to the Slovak Government Regulation GR 395/2006 relating to PPE.

**Table 3.** Hazards that PPE protects from as follows.

| Hazards | Causes and Types of Hazards | Criteria and Security Measures, Making Personal Protective Equipment |
|---|---|---|
| Mechanical | - objects falling on the front of the boot or its compression<br>- fall and heel strikes<br>- fall due to slipping<br>- stepping on sharp or pointed objects<br>- effects on individual parts of the leg:<br>• shinbone<br>• ankle<br>• instep<br>• foot | - resistance of the vamp (toe scuff)<br>- heel's ability to absorb the impact<br>- strengthening the instep<br>- anti-slippery properties<br>- outsole resistant to perforation, cut, puncture<br>- effective protection of individual parts of feet |
| Electrical | - low and medium voltage<br>- static electricity | - electrical insulation<br>- discharge of static charge |
| Thermal | - heat or cold<br>- spraying of molten metal | - thermal insulation against heat or cold<br>- impermeability, resistance |
| Chemical | - harmful or dangerous fluids or dust | - impermeability, resistance |
| Excessive load | - excessive standing or walking | - moulding the tread of footwear |

**Table 3.** *Cont.*

| PPE shortcomings | Causes and types of shortcomings | Criteria and security measures, making personal protective equipment |
|---|---|---|
| **Discomfort, limitation, interference with work** | - insufficient comfort in use:<br>• non-adjustment of footwear to feet<br>• insufficient sweat absorption<br>• fatigue caused by wearing foot wear<br>• water permeability | - ergonomic solution (structure):<br>• shape, size and boot insole,<br>• water vapour permeability and ability to absorb water vapour<br>• flexibility, weight<br>• waterproofness |
| **General** | - insufficient protection against health risks<br>- insufficient cleanliness<br>- dangers of sprains and leg dislocations | - properties (material quality)<br>- easy maintenance and care<br>- rigidness of footwear in transverse direction, arch support, and suitability of footwear |
| **Alteration of protective functions due to aging** | - influence of weather and surrounding conditions, cleaning, wearing | - resistance to operational wear and tear<br>- preservation of protective functions throughout the whole time of use<br>- resistance of outsole to corrosion, abrasion and stress |
| **Static electricity** | - electrostatic discharge | - electrical conductivity |
| **Insufficient protective effect** | - wrong choice of personal protective work equipment | - choice of personal protective work equipment according to the type of hazards, level of risk and extent of work:<br>• follow manufacturer's instructions (user instruction)<br>• compliance with labels and symbols on personal protective work equipment, e.g., level of protection, special use<br>- choice of personal protective work equipment after considering individual factors such as size, weight and fit of the user. |
| | - improper use of personal protective work equipment | - proper use of personal protective work equipment, knowledge and awareness of hazards and risks<br>- respecting the manufacturer's instructions |

Figure 8 presents the results of research performed by the NFPA (National Fire Protecting Organisation) in 2016–2020. These results are summarized in the form of graph describing percentage occurrence of fires and fire-related injuries according to the individual months within a year [29]. This distribution reflects the impact of cold and hot weather conditions on firefighter work associated with application of personal protective equipment. It is visible that the monthly share of accidents in December, January and February was somewhat lower compared to the distribution of fires in most of the remaining months. The firefighters are regularly exposed to risk of injury during their working activities, which also include exposure to fire or explosion hazards and the carrying of heavy equipment or the injured people. They also struggle with hot or cold weather and with slippery or uneven surface conditions, where usage of the proper protective footwear is important. The next graph in Figure 9, which originates from the same research source, illustrates a dependence of percentage occurrence of fires and fire-related injuries on daily time factor. It is evident that the critical time with the highest frequency of firefighters' injury occurrence is interval between 12 p.m. and 8 a.m. due to firefighter interventions at night or in dark conditions [29].

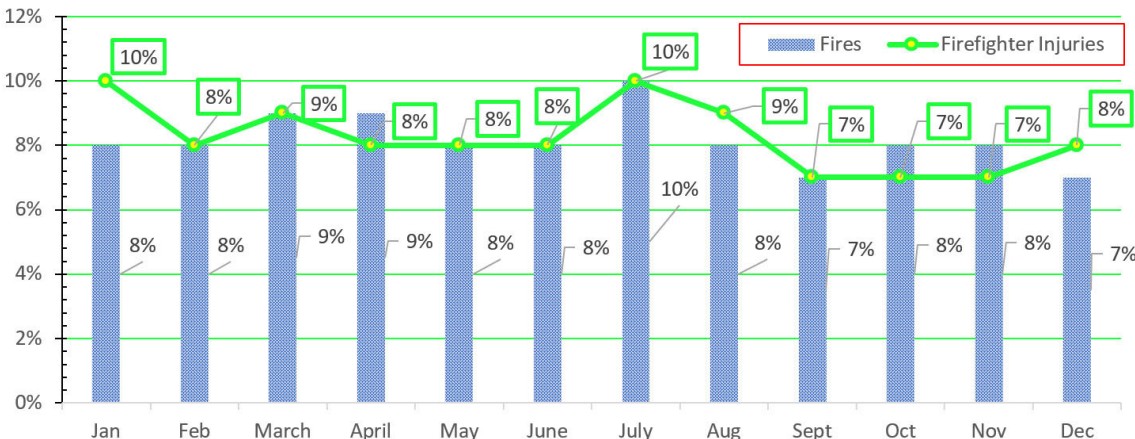

**Figure 8.** Percentage occurrence of fires and fire-related injuries according to months in 2016–2020.

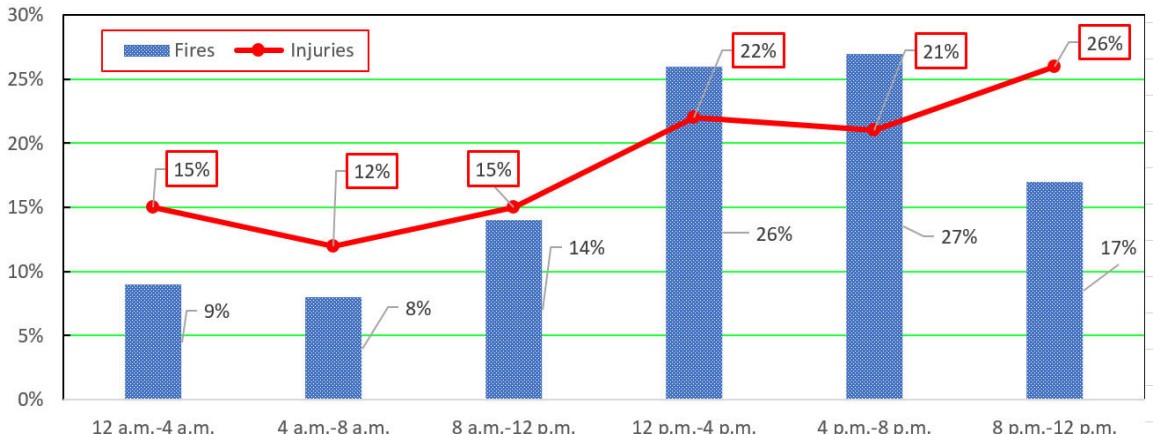

**Figure 9.** Percentage occurrence of fires and fire-related injuries depending on daily time factor (e.g., intervention in dark space or during night) in 2016–2020.

*6.2. Firefighting Boots*

Firefighting boots are manufactured from these materials:

- Hydrophobic waterproof leather;
- Lining laminate equipped with special foam interior is extra resistant to wear; the leather upper is covered by quality protective rubber;
- Rubber cap is placed in toe cap area.

The process of firefighting boot production has to fulfil the highest standards of quality. This production process is certified according to the EN 15090:2012: The current effective version of the European standard valid for firefighting safety boots. F2A: All the firefighting interventions, as well as the rescue operations, require a high level of toe protection and protection against penetration, including the antistatic protection. HI3: Maximal thermal insulation, CI: Maximal cold insulation, AN: Ankle protection, SRC: Anti-slip class C (the highest level of protection). Figure 10 introduces, as an illustrative example, a special easy to use system of putting on and taking off for the firefighter boot [43].

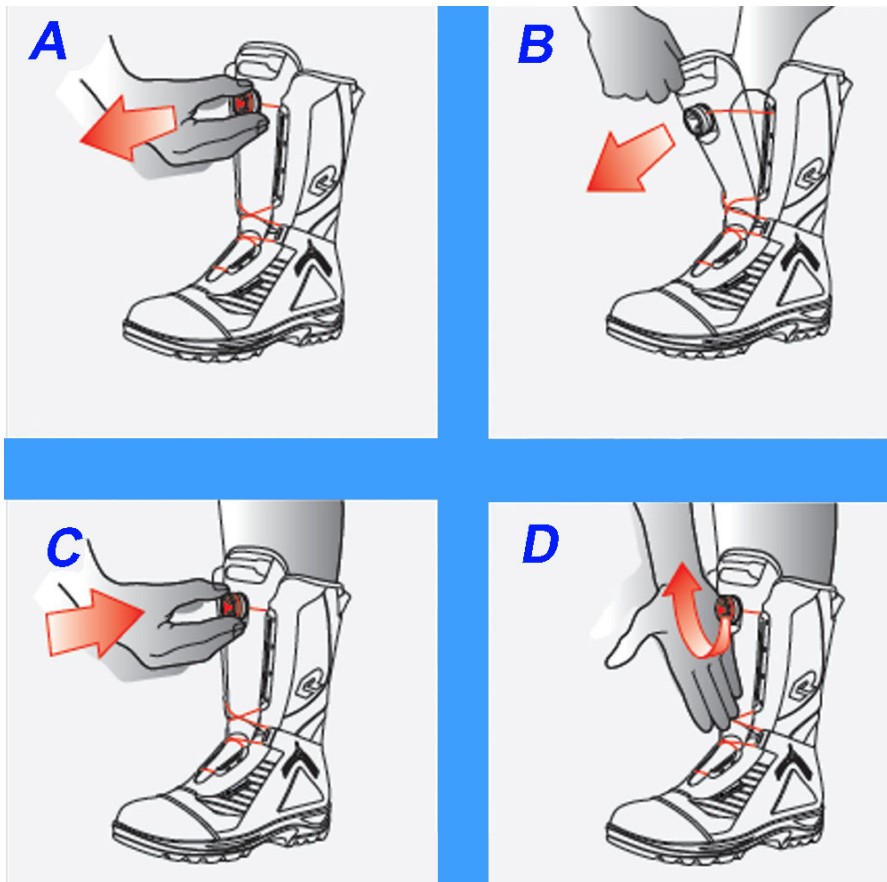

**Figure 10.** Easy to use system of putting on and taking off—Rosenbauer Firefighting Boots—for firefighting and rescue operations. Explanatory notes: (**A**)—Pull on the Boa Dial, (**B**)—Pull the leather tongue forward and step into the boot, (**C**)—Push in the Boa Dial, (**D**)—Turn the Boa Dial with the edge of your hand until the boot fits perfectly.

## 7. Example of Footwear Determined for Firefighting Intervention

*7.1. The Footwear for Firefighting Intervention in the Case of Forest Fires VFT XTREME BOOTS—EN 15090:2012 F1PA + HI3 + CI + SRC, Oeko-Tex*

The footwear determined for firefighting intervention in the case of forest fires are characterised by the following properties, Figure 11 [44]:

- it is a special piece of footwear determined for firefighting intervention in the case of field fires and forest fires;
- it stands out for its low weight, extreme comfort, top protection and high durability;
- it is designed for long-term firefighting interventions in difficult terrain;
- it contains innovative materials and structures;
- it is certified according to standards valid for firefighting footwear;
- it is fire resistant, glow resistant, waterproof and antistatic;
- the upper part of the shoe is made from a light fabric, which is resistant to mechanical wear and damage;
- the shoes are fastened by quick-release fasteners and non-flammable laces;
- it contains Kevlar protection against puncture;
- is equipped with a protection of the front part, which is made from durable microfibers, against impact of heavy objects and against digging;
- it has a nitrile-rubber sole;
- it is ecological, recyclable, antibacterial and antifungal.

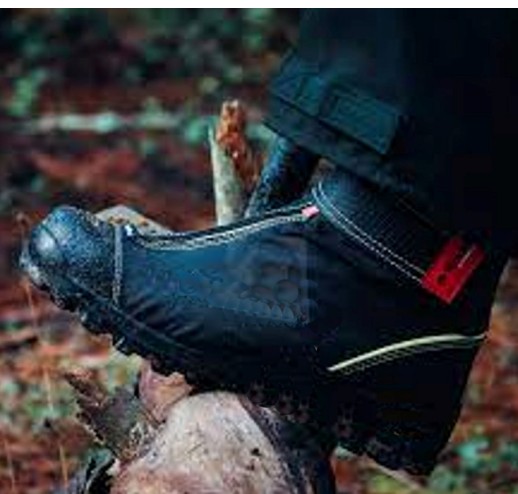

**Figure 11.** Firefighter boots used in forest fires.

*7.2. Special Functional Elements on Firefighter Shoes*

In order for the firefighting footwear to be maintained and in operation for a long time, it must be constantly improved and equipped with modern elements to ensure its proper functionality and durability. It is necessary to use suitable materials, a quality design and special improvements. According to experiences, rubber as the basic material does not appear to be the best solution. As illustrated in Figure 12, rubber shoes often wear out. The ideal material is leather [33].

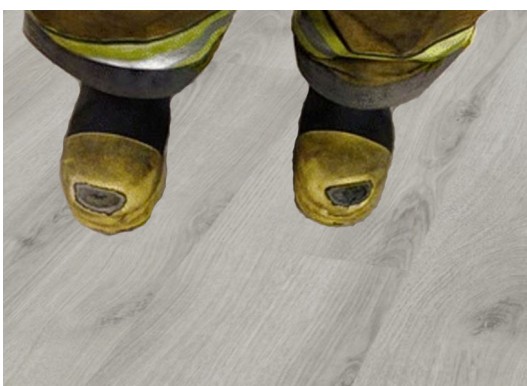

**Figure 12.** Worn out toe cap due to frequent friction on the ground while crawling.

Also important is a suitable arrangement of the lower surface on work shoes with regard to high efficiency in terrain. Figure 13 shows an example of logical arrangement of the protrusions and their role according to the colour markings in this figure [45]. All protrusions together form the overall pattern design, Figure 14 [45].

A proper sole should meet the following characteristics:

- wide contact areas determined for grip on rocky surfaces;
- placement of areas in lines for natural bend and flexibility;
- slots in the ankle area help in uneven terrain;
- the overall pattern has a medium depth to be maximally functional on various surfaces.

Nowadays, there is a special emphasis placed on assortment in the area of work and firefighting footwear, where the sole must pass demanding tests for anti-slip properties (trekking footwear, military and police footwear), for resistance to oils and fuels (firefighter footwear) or must meet the requirement for a low rate of wear (forest worker and hunter shoes). Extraordinary high-quality material mixtures ensure that the sole does not change its shape in contact with high temperatures, or that it does not crack in extreme cold.

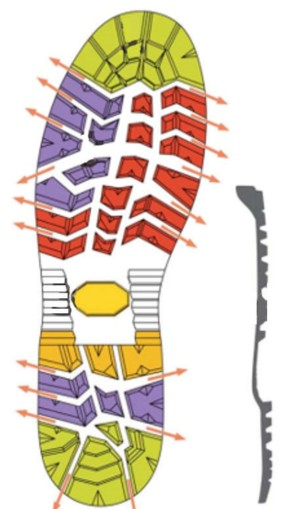

Green—maximum adhesion on a smooth surface.

Red—ensures good forward traction when starting and walking in a soft terrain.

Orange—braking, helps when walking downhill on a soft or hard terrain.

Violet—the shape and placement of these protrusions improves overall stability.

Orange arrows—indicate "open" pattern, which improves self-cleaning effect. This means that dirt does not stick to the pattern and easily falls out when walking.

**Figure 13.** Example of logical arrangement of protrusions on the sole.

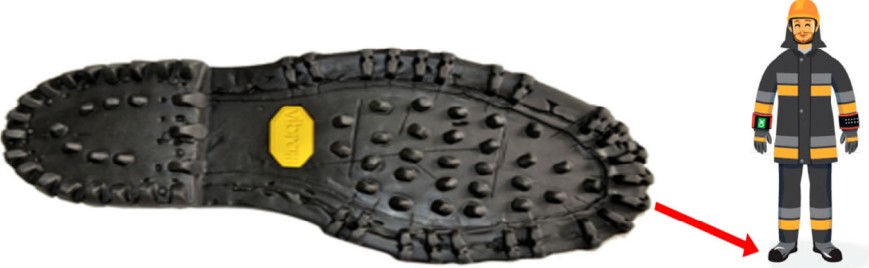

**Figure 14.** The sole design created by VIBRAM® is protected by patent and copyright.

These several types of materials are combined with many patterns design. The pattern is not random at all and created only for the aesthetic side. Whether it is a sole for trekking or work shoes, the tread design must meet several conditions.

### 7.3. The Footwear for Firefighting Intervention of Type TORNADO

The TORNADO type firefighting intervention shoes (firefighter boots) are a combination of non-slip lacing shoes with a zipper, which has proven to be an excellent solution. This kind of shoe is equipped with anti-cut protection. It contains additional layers made from Kevlar, which are situated between top cover leather and special membrane Sympatex. This footwear is manufactured in accordance with the European standard EN 381 [46], Part 3—Protective Equipment for User Working with a Chainsaw when Removing Forest Vegetation during Fires—Class 2. The Tornado footwear is certified according to the European standard EN 15090:2012 F2A HI3 CI AN SRC. In addition, they also met the requirements for a high class of resistance to radiant heat, for insulation against the cold, for ankle protection and for comfortable wear. All this was achieved thanks to four flexible zones and equipment with a Sympatex membrane. These shoes can be laced at two different levels: around the calf and around the instep. The mentioned two lacing zones are separated from each other by a special stopper situated in the ankle area, which allows a different level of tightening of the laces in each zone. No other system allows such an individual adjustment of footwear for every foot shape. Just one shoe adjustment needs to be made by the user and the system will stably maintain the optimal level of lacing in each zone. The zipper makes it easy to take off and put on shoes. Protection of the ankle is provided by padding added in the ankle area in order to reach maximum protection against pressure and impacts. These paddings were tested according to the standard EN 20345 [47]. Thanks to this footwear solution, an increased protection is also provided against ankle sprains and against the risk of penetration by sharp objects. This special footwear is breathable

and waterproof too. It also provides protection against viruses, bacteria, body fluids and various chemicals. The necessary high comfort during activities such as climbing, kneeling and driving is achieved thanks to the above-mentioned four flexible zones located above the heel, in the area of the ankle joint and in the area of the calf. An example of such footwear can be seen in Figures 11 and 15 [44].

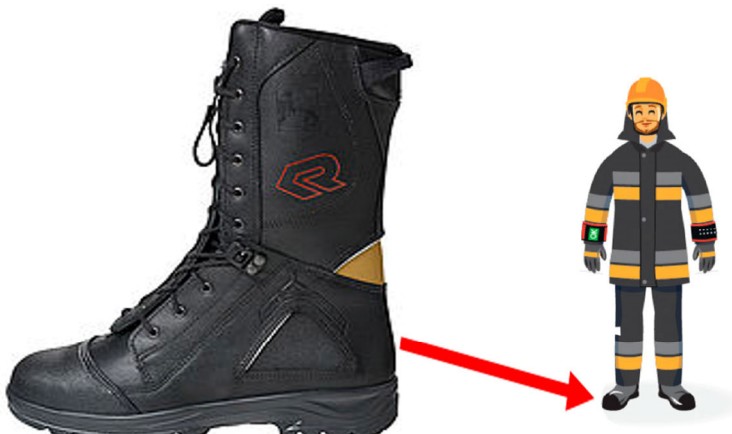

**Figure 15.** Firefighter boots TORNADO.

## 8. Research Advances in Additive Manufacturing and in Sustainable Industrial Engineering

Currently, the additive technologies are also used in the production of firefighting footwear. These digital engineering technologies belong to the category of advanced industrial engineering within the prospective technologies. A set of advanced industrial engineering techniques and procedures creates a system, which is collectively called Digital Enterprise. This development trend is the result of incremental development of production system functionality as a necessary response to changes happening now in human civilization, especially to the application of new technologies. The changes are mainly concentrated in the area of information and communication technologies. The above-mentioned term increments are part of the overall functionality of the product. The additive production is characterised by the fact that, during the development phase, the requirements for the product as a whole are first described and then the individual product parts are implemented. This article is focused on the category of firefighter personal protective equipment, which is determined for the safe and reliable activity of firefighters during their intervention, whereby the chosen specific product is the firefighters' footwear. The most important requirements concerning the firefighting footwear, their development and testing are specified in this article. The process of product development is finished only when the overall required functionality of the product is achieved [48].

If the basic requirement is to dispose of a sustainable competitive advantage, it is necessary to apply innovative methods of industrial engineering using the new advanced technologies, in close integration with the information and communication technologies. Within the dynamically developing global competition, relating to the industrial products, the most important ability of the industrial enterprises is to meet, as fast as possible, the demands, needs, wishes and possibilities of customers in various structures. Not only are the delivery terms of the finished products to the customer shortened, but so are the product innovation cycles with the aim to commercialise the achieved research results as soon as possible. These practical goals are increasingly linked to the use of information and communication technologies [49].

Since the competitive environment is still risky, dynamic and turbulent, the production companies must react to this fact by the innovation of their processes, working methods and organizational structure. Those industrial companies which do not undertake such innovations are unlikely to survive. Producers of the protective firefighting footwear

also identify with these demanding requirements. Before preparing the production itself, it is necessary to define the area of use for the footwear, the expected characteristics of such footwear, where the footwear production will be oriented and which benefit the given footwear offers. Answers to these questions depend primarily on the firefighter's requirements. Sometimes it is enough to upgrade what already exists due to outdated design of a given product or material changes. After determining and evaluating the target user requirements, working meetings in design studios and internal brainstorming can follow. Consequently, creation of the product prototype is based on this information. The produced prototype is evaluated with regard to comfort, durability and overall design of final product. A typical example of such production procedure result is the special, top-quality firefighter footwear, presented in Figure 16.

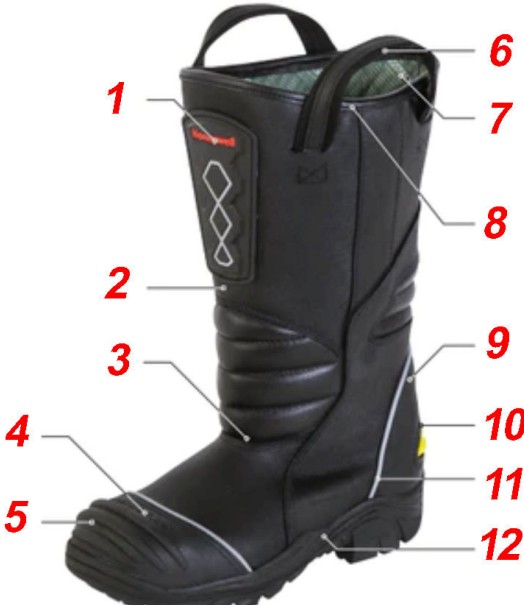

**Figure 16.** Firefighter's footwear NightHawk™ 5555 PRO Series from Honeywell. Explanatory notes: 1. Rubber Shin Guard, 2. Heavyweight Kevlar, 3. Premium Full-Grain Cowhide Leather, 4. Power Toe Protection, 5. Oblique Steel Toe Cap, 6. Leather-Reinforced Pull Straps, 7. Footwear Fabric Lining, 8. Leather Top Binding, 9. Heel and Ankle Grip System, 10. Heel Kicker, 11. Reflective Trim, 12. Rugged Reliable Protection.

Nowadays, the preferred producers of firefighting footwear are those that support the latest research results and advances reached in the additive manufacturing of this footwear, offer the most modern design as well as apply improvements for higher reliability and, above all, durability of the firefighting footwear. One of such top producers is the company Honeywell [50]. An application example of the above-mentioned features is the PRO Series Nighthawk 5555 Firefighting Boots, Figure 16. Integrated into these innovative boots are the most modern footwear elements of today, namely:

1. Rubber Shin Guard—Ladder protection with cushioned backing for comfort and convenience;
2. Heavyweight Kevlar—Premium thread to maximize seam length and durability.
3. Premium Full-Grain Cowhide Leather—Flame-resistant, water-resistant, leather upper, flexible joint transitions for world-class comfort;
4. Power Toe Protection—Updated innovation for outstanding abrasion resistant durability;
5. Oblique Steel Toe Cap—A roomy fit without unnecessary bulk;
6. Leather-Reinforced Pull Straps—Easy grip and kind to pants and legs;

7. Footwear Fabric Lining—Full-height protection and comfort from underfoot to the boot-top line. Waterproof and blood borne pathogen resistant;
8. Leather Top Binding—Always a smooth, finished interface for pants and legs;
9. Heel and Ankle Grip System—Moulded heel counter, contoured, lined, integrated cushion support for outstanding fit and comfort;
10. Heel Kicker—High visibility doffing aid;
11. Reflective Trim—Located below the pant line for maximum benefit. Cleanable and heat resistant;
12. Rugged Reliable Protection—Steel triple-ladder shank and steel flex-sole plate. Puncture and penetration resistance.

The description of this footwear is a clear demonstration of how the most modern design elements are related to research and progress in the field of additive manufacturing. These shoes are manufactured from the top materials that are strong, durable and also lightweight. Another positive aspect is the utilisation of leather, which is an inexhaustible and long-term sustainable material resource. Artificial leather can be used as a replacement for genuine leather with regard to the fact that the artificial leather already has properties comparable to genuine leather. This fact also contributes to social and environmental progress. Utilisation of these materials significantly increases the durability of firefighter footwear, as well as the comfort and safety of firefighters at fire extinguishing intervention.

However, it is important to mention the fact that the service life of firefighting footwear depends on where it is used, on the intensity of use, on compliance with the recommendations from producer as well as in a significant extent on testing procedures described in this article. Today, the smart and flexible production lines are used in production of firefighter footwear.

The modern shoe production line is a system of smart innovative equipment integrated within the entire industrial chain. This system is specially developed for shoe manufacturing companies, whereby it enables them to create a flexible and smart production line. The smart shoe production line integrates robots, robotic 3D vision systems, large amount of processed data and artificial intelligence technologies. In this way, there is a transformation and modernisation of the traditional shoe industry towards smart production. The up-to-date production systems increase the level of production automation, reduce production costs and improve production efficiency and product quality within the traditional shoe industry as well as enabling us to realise automated shoe production based on human–machine cooperation [51].

The described additive production, which utilises the smart and flexible production line, is a key technology for sustainable development and smart footwear solutions. All these above-mentioned aspects emphasize relevance of the research advances in additive manufacturing and sustainable industrial engineering [52].

## 9. Conclusions

Firefighting equipment includes protective safety boots, which make it possible for firefighters to enter a potentially dangerous environment relatively safely. This article underlines the importance of firefighter boots with regard to various operational safety factors. In addition to high temperatures, the firefighters must deal with other hazards at work such as slippery surfaces, rough terrain, flooded areas, sharp objects, etc. These details are clearly visible in hard-to-reach terrain such as a forest. Firefighting boots ensure proper fit, traction abilities and flexible movement for the firefighters and also for other fire safety experts. They are made from materials that give high levels of resistance and protection for firefighters in rescue operations. In addition to sprains and strains or exhaustion-related injuries, firefighters experience a lot of traumatic injuries including cuts and lacerations, burns, fractures, etc. Personal protective equipment, training and certified protective equipment are essential for the protection of firefighters against such injuries.

If the National Fire Protection Association NFPA [33] standard for firefighting boots gives any indication, there are requirements and attributes set for protective firefighting

boots. The sole must be melt-resistant, able to sustainably operate in hot environments without melting, softening or otherwise being damaged. The sole must also offer a great deal of traction and should be of a lug sole design. Firefighting boots must be tall to give the wearer maximum support in the heel and ankle areas. They should provide firm footing and stability when walking around fire-damaged areas. Plastic and rubber become a liquid at relatively moderate temperatures, so the material that is most suitable for firefighting boots is leather. The stronger and the higher the quality of the leather, the better.

The main role of the personal protective equipment determined for the firefighters is not only to protect them, but also to allow them to perform a safe intervention in emergency situations. They serve primarily to enable firefighters to handle a dangerous situation without injury. Therefore, it is necessary to assess the risks, to identify the threats and to evaluate the specific requirements for such properties of the personal protective equipment that will reduce the risks or eliminate them.

Hot, cold and also humid conditions affect the protection of the firefighter at the place of the fire. Thermal, physical and other hazards must also be taken into account when determining risks. The type, degree and time of exposure to heat, as well as the physical environment, have a significant impact on the potential hazards the firefighters face. Hazards such as contact with flame, low visibility, chemicals and uneven surfaces are also significant because they increase the probability of injury during intervention.

Although firefighter footwear is designed to prevent injury, the limitations of such footwear in providing protection must be respected. The protective effect of the footwear may be limited due to some design elements or material properties, as well as due to the weight of the footwear. Footwear should fit properly to provide the firefighter with the necessary protection.

Intelligent personal protective equipment, which represents intelligent protection of the future, is also very important.

Intelligent personal protective equipment (PPE) is occurring increasingly often; it is presented at professional trade fairs and has been in use for some time. However, current applications of such products are still somewhat problematic. Although some quality products of such kind already exist, smart PPE is a rapidly developing field and all participants are still learning how to fully utilize the potential of smart PPE. The level of protection can be increased either by using improved materials or by electronic components in PPE. The improved materials are characterised by the new quality properties. For example, the knee pads are often inflexible and they prevent normal movement. On the other side, the intelligent shock-absorbing material can be both soft and flexible and it allows normal movement. However, if protection is required in the case of an impact, the properties of the smart material will be suitably changed and the effect of shock absorption is manifested.

The benefit of this article is that it points out the importance of firefighter footwear testing, which has a significant influence on health protection for firefighters during their demanding interventions. Firefighters' shoes should also belong to intelligent PPE. The firefighter profession requires a high level of physical fitness, speed and endurance, whereby the footwear is an essential part of the equipment. The proper footwear provides advantages ranging from prevention of foot injuries to improvement of overall performance at work. The firefighter footwear has to fulfil the following Top 5 functions, which are presented in Figure 17.

Finally, it is necessary to emphasize the fact that in order for the firefighting equipment to be sustainable in today's conditions and fires, it is important to always use the highest quality materials. In addition, it is also required that we apply the appropriate testing methods determined for control and the new standards that prescribe manufacturing and testing methods. Leather is the best material for firefighter footwear (Table 1—Class I). As far as sustainable design is concerned, it is necessary to implement all the improvements and innovations mentioned in this article that are certified by the highest ISO quality standard.

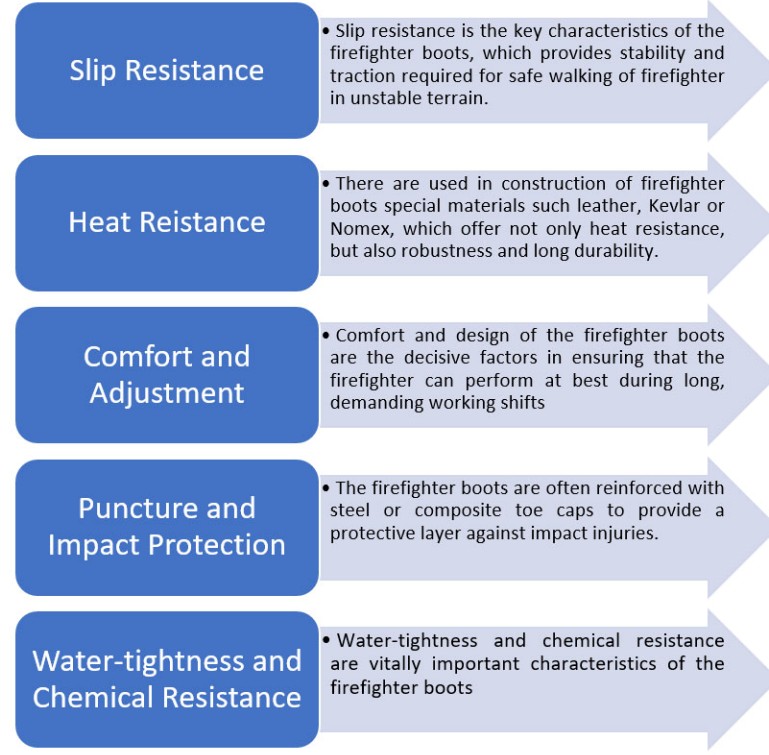

**Figure 17.** The Top 5 functions of the firefighter footwear.

**Author Contributions:** M.T. performed the professional analyses and wrote the manuscript. J.K. controlled the work, and he managed the writing and reworking of previous drafts, including the final approval. J.K. collected and prepared the obtained data. All authors have read and agreed to the published version of the manuscript.

**Funding:** This research was supported by APVV project No. 19-0367, KEGA 013 TUKE-4/2020, KEGA, 029TUKE-4/2021.

**Institutional Review Board Statement:** Not applicable.

**Informed Consent Statement:** Not applicable.

**Data Availability Statement:** Not applicable.

**Conflicts of Interest:** The authors declare no conflict of interest.

## Abbreviations

| | |
|---|---|
| PPE | Personal protective equipment |
| IPCC | Intergovernmental Panel on Climate Change |
| STN | Slovak technical standard |
| EN | European standard |

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
