# Peer review of "Features and Sustainable Design of Firefighting Safety Footwear for Fire Extinguishing and Rescue Operations"

_sustainability, doi:10.3390/su152015108_

Round 1

Reviewer 1 Report

1. Generally, the part of introduction would focus on the main topic with concise backgrounds. However, this paper lists tedious content about the analysis of the importance and emergence of protection of forest firefighter, while the research theme firefighting boots was introduced with tiny words.

2. In Chapter 3, the test method of fire fighting shoes would be the key part of evaluation and optimaise of its safety performance. Therefore, only with the discuss of flame retardant performance and zipper test methods, the verification of high quality design of fire fighting shoes is unconvinced. It’s reccomended to refer to sufficient testing or evaluating methods, such as mechanical properties and sole anti-slip properties to reach a comprehensive analysis.

3. Specifically, the innovation is unclear and insufficient. It’s understandable to list the existing standards and test methods, however with lacks of summarise and personnal thinking, the new ideas of this paper was less preminent.

4. On the whole, the logical framework and content coherence should be strengthen with focusing on the main point of this study and weakening the existing investigation or previous researches.

Author Response

Thank you very much  for reading of our article and for your valuable comments. We corrected our article according to your comments and also according to requirements of the Academic Editor.  We hope, we have met the requirements.

Answer 1.

We corrected and supplemented the introductory and conclusion part of the article. The missing explanatory text parts are supplemented; some descriptions of the images are clarified. In the text section, we have added the top features of the firefighting footwear, which are the most important from the sustainability point of view. The main goal and contribution of the article is declared. The article is modified and the cited references are changed due to reduction of the similarity index value.

Answer 2.

Within the subchapter 3.1, we added explanation regarding firefighter footwear tests. The EN ISO 15025:2002 standard specifies the minimum requirements and test methods intended for three types of the footwear for firefighters, namely the firefighter footwear determined for the common rescue operations, for the fire rescue interventions and for the rescue actions during accidents with hazardous materials. Other footwear tests are also listed in the given standard, but we selected only several of them into our article. The pressure resistance test of the toe cap is performed according to the EN ISO 20344:2011 standard. There are defined requirements to the protective toe cap concerning the internal length of the reinforcement, resistance to impact and pressure, and also resistance of the metal reinforcement against corrosion.

Answer 3.

We added into the final chapter of the article new information about the intelligent personal protective equipment as an integrated part of the actual development trend related to implementation of the intelligent protection in the future. We also presented the so-called top features of the firefighter footwear, which are the most important in terms of sustainability of this footwear in usage. We emphasized contribution of the article in the given subject area.

Answer 4.

We added additional desgnation of the firefighter footwear. There is also added the Table 2. Description and explanation to Figure 7, 8 and 9 is supplemented. The new Figure 16 is important with regard to sustainability of the firefighting footwear in use. The main goal and contribution of this article is emphasized in Introduction and Conclusion.

Reviewer 2 Report

Overall, original and interesting paper. I would suggest considering early clarification in the paper distinguishing what type of firefighter you are speaking about; for example, the emphasis of the body seems to focus on wildland however there is reference to structural firefighters as well. It may be helpful for the reader if you specify which type you are writing about. 

Consider if you want to change the spelling of exposure in line 11 to eliminate the hyphen. In the US, it is unusual to hyphenate this word. 

Author Response

Thank you very much  for reading of our article and for your valuable comments. We corrected our article according to your comments and also according to requirements of the Academic Editor.  We hope, we have met the requirements.

Answer:

We corrected and supplemented the introductory and conclusion part of the article. The missing explanatory text parts are supplemented; some descriptions of the images are clarified. In the text section, we have added the top features of the firefighting footwear, which are the most important from the sustainability point of view. The main goal and contribution of the article is declared. The article is modified and the cited references are changed due to reduction of the similarity index value.

There is added to the article an information that this article is focused on such firefighter footwear, which is determined primarily for the firefighters working outdoors, especially in the forest, in difficult conditions

Reviewer 3 Report

Firefighters are exposed to complex and changing environments when carrying out firefighting and rescue work, and personal protection is very important. This manuscript presents new ideas, concepts, recent advances and technical tools for a sustainable use of firefighting boots. This may be useful to improve the safety of firefighting boots. However, the manuscript has many structural problems and some of the studies lack rigour. Therefore, we recommend a major revision. The details are as follows:

1.In lines 32-33, there seems to be some problems with the expression of the year, and the author is advised to check the numerical format and units of the manuscript.

2. Authors spend a great deal of their abstracts presenting the background of their research, and in order to give the reader a quick overview of your contribution, the research contribution and novelty of the paper should be more clearly expressed.

3. In lines 124-132, the author describes the effects of noise and vibration from firefighting equipment on firefighters, which seems to have little to do with the subject matter of the manuscript and may be logically problematic.

4. It is recommended that the authors recheck and reorganise the paragraphs and logic of the introduction.

5. The title of figure 8 does not seem to convey the same thing as the picture. Why is the number of fires in Figures 8 and 9 expressed as a percentage? What does this mean? In addition, the title of 6.1 is "Criteria for the selection of protective footwear", and Figures 8 and 9 do not seem to have much relevance to that title.

6. Figures 7, 8, and 9 in the manuscript do not have corresponding descriptions, and the authors are advised to add explanations and elaborations of the images.

7. It does not seem reasonable that the authors have included "Marking of Fire Fighting Footwear" as a separate chapter 4, with only one diagram and a short description.

8. The authors mention five tests for firefighting footwear in Chapter 3, but 3.1 and 3.2 only mention "Testing of footwear for flame resistance" and "Test of zipper" without any description of "Pressure resistance of toe cap test" and "Insulation against heat test", which may make the thesis lack of rigour.

9. In the conclusion, it is recommended that the authors summarise the key results in a concise form and highlight their significance.

10. The authors present new ideas and concepts for sustainable design of firefighting boots. We suggest the authors to compare it with previous studies and indicate its advantages to reflect the novelty of the study.

11. The authors may need to improve the language and narration of the article to ensure that the article's content is more concise and clearer to make it more acceptable to readers.

The authors may need to improve the language and narration of the article to ensure that the article's content is more concise and clearer to make it more acceptable to readers.

Author Response

Thank you very much  for reading of our article and for your valuable comments. We corrected our article according to your comments and also according to requirements of the Academic Editor.  We hope, we have met the requirements.

Answer 1.

We corrected information about the number of years exactly according to the literature source. We used a new, more general formulation: “during the time period of future several tens years”, as it is indicated by the authors from the source [1.2].

Answer 2.

We modified the chapter Introduction and removed some less relevant parts in order to formulate the introductory information more clearly for the reader, with regard to a better understanding of it. In this way, the Introduction is reorganised as well as simplified.  

Answer 3.

We agree with the reviewer’s comment. Noise and vibrations should not be included in this article, because it is focused on the footwear and safety of firefighters, what is not related to the noise and vibrations. Therefore, we removed this part.

Answer 4.

We modified the chapter Introduction and removed some less relevant parts in order to formulate the introductory information more clearly for the reader, with regard to a better understanding of it. In this way, the Introduction is reorganised as well as simplified.  

Answer 5.

We added, respectively modified the explanatory descriptions to Figure 8 and 9. We included the explanation of the graphs into the text. Expression in percentages in Fig. 8 and 9 is according to the National Fire Protection Association (NFPA), where the number of injuries in the graphic outputs is presented in percentages.

Answer 6.

We supplemented descriptions to Figures 7, 8 and 9 as well as we described in the text content and meaning of these figures.

Answer 7.

We presented additional designation of the firefighter footwear. There is supplemented the Table 2.

Answer 8.

Within the subchapter 3.1, we added explanation regarding firefighter footwear tests. The EN ISO 15025:2002 standard specifies the minimum requirements and test methods intended for three types of the footwear for firefighters, namely the firefighter footwear determined for the common rescue operations, for the fire rescue interventions and for the rescue actions during accidents with hazardous materials. Other footwear tests are also listed in the given standard, but we selected only several of them into our article. The pressure resistance test of the toe cap is performed according to the EN ISO 20344:2011 standard. There are defined requirements to the protective toe cap concerning the internal length of the reinforcement, resistance to impact and pressure, and also resistance of the metal reinforcement against corrosion.

Answer 9.

We modified and supplemented the Conclusion chapter, in which we underlined the specific role and importance of the personal protective equipment. There is also emphasized in this part a contribution of the article to the given topic.

Answer 10.

We added into the final chapter of the article new information about the intelligent personal protective equipment as an integrated part of the actual development trend related to implementation of the intelligent protection in the future. We also presented the so-called top features of the firefighter footwear, which are the most important in terms of sustainability of this footwear in usage. We emphasized contribution of the article in the given subject area.

Answer 11.

We corrected and supplemented the introductory and conclusion part of the article. The missing explanatory text parts are supplemented; some descriptions of the images are clarified. In the text section, we have added the top features of the firefighting footwear, which are the most important from the sustainability point of view. The main goal and contribution of the article is declared. The article is modified and the cited references are changed due to reduction of the similarity index value.

Round 2

Reviewer 1 Report

The revised content of this paper have met the requirements.

Author Response

Dear reviewer, thank you very much for your second revision of our manuscript. We performed the required minor revisions in order to make another improvement of our article. Namelly, we improved explanation of the figures and tables as well as their interconnection with the text. We also underlines results and benefit of this our article.

Reviewer 3 Report

The authors have carefully revised the manuscript. I agree to accept it.

Author Response

(The authors gave the same response as above.)
